# Effect of Titanium Addition on As-Cast Structure and High-Temperature Tensile Property of 20Cr-8Ni Stainless Steel for Heavy Castings

**Qiming Wang, Guoguang Cheng \* and Yuyang Hou**

State Key Laboratory of Advanced Metallurgy, University of Science and Technology Beijing, Beijing 100083, China; wangqiming_1310@sina.com (Q.W.); hyyustb@gmail.com (Y.H.)
\* Correspondence: chengguoguang@metall.ustb.edu.cn; Tel.: +86-010-6233-4664

**Abstract:** 20Cr-8Ni stainless steel is used to manufacture heavy castings in industrial practice. Owing to the slow solidification rate of heavy castings, the as-cast structure is usually coarse, which reduces the mechanical properties. To refine the solidification structure, the effect of titanium addition on the as-cast structure and high-temperature tensile property was investigated in the laboratory. The ingots with 0.0036, 0.2, and 0.45 mass percent titanium were produced in the laboratory. On the basis of experiment and thermodynamic calculation through Thermo-Calc software, the typical inclusions changed from dual phase Si-Mn-Ti oxides to pure TiN or complex $Ti_2O_3$ + TiN with titanium content increasing. The equiaxed grain ratio of ingot increased from 51 percent to nearly 100 percent, and the size of equiaxed grain decreased owing to heterogeneous nucleation of δ-Fe. Besides, the equilibrium solidification model changed from the FA model to the F model, and the mass fraction of ferritic phase in ingots increased from 24.8 to 42.6 percent. As a result, the yield strength and ultimate tensile strength of ingots increased gradually, but the tensile elongation changed little owing to the increase of ferritic phase mass fraction and decrease of equiaxed grain size.

**Keywords:** stainless steel; as-cast structure; microstructure; tensile property; titanium content

## 1. Introduction

20Cr-8Ni stainless steel is used to manufacture heavy castings owing to its perfect corrosion resistance and mechanical properties [1,2], such as pump casing for nuclear power. However, owing to the slow solidification rate of heavy castings, the as-cast structure is usually coarse, which reduces the mechanical properties. Nitrogen is usually used to improve mechanical properties owing to the intrinsic solid solution hardening effect [3,4]. However, nitrogen gas pores are usually formed during the casting process of heavy castings [5,6]. Therefore, it is necessary to refine the as-cast structure of heavy castings, and titanium addition is a common method to refine the solidification structure.

It is well known that titanium addition in steel would cause inclusion problems. Oxides containing titanium are the typical inclusions formed in steel firstly [7–11]. Seo et al. concluded that the typical inclusions changed from $MnSiO_3$ inclusions, (Mn, Ti)-spinel oxide, to $Ti_2O_3$ with titanium content increasing in Ti-containing steel weld metals [12]. Besides oxides rich in titanium, titanium nitride is the other typical inclusions in titanium-contained steel. Ozturk et al. investigated the thermodynamics of titanium in Fe-Cr alloys and formation of TiN in Fe-Cr-N-Ti alloys [13]. Yin et al. found pure TiN particles and complex TiN with $Al_2O_3$-MgO-$TiO_x$ cores in Ti-stabilized 17Cr austenitic stainless steel. Stringer-shaped TiN inclusions were also observed in the rolled bar [14].

On the basis of oxide metallurgy, the oxides rich in titanium and TiN could refine the solidification structure through heterogeneous nucleation [15–17]. Bramfitt found the misfit factor between TiN and

δ-Fe was small, and TiN was very effective as a nucleating agent for δ-Fe [18]. Park found that TiN and MgAl$_2$O$_4$-TiN complex inclusions promoted the formation of fine equiaxed grains when Ti was added in ferritic stainless steel [19]. Nakajima et al. studied TiN, Al$_2$O$_3$, and Ti$_2$O$_3$ on heterogeneous nucleation in pure Fe and Fe-Ni alloys. They found that only TiN had a practical effect as a catalyst on the triggered nucleation of the primary crystal of δ phase, but none of them had a practical effect on the nucleation of the primary crystal of γ phase [20]. They also investigated TiN and Al$_2$O$_3$ on heterogeneous nucleation in pure Fe-Ni-Cr alloys, and TiN was the greater catalyst compared with Al$_2$O$_3$ [21]. It is known that mechanical properties could be improved through refining grains of steel [22–24]. Schino et al. investigated the effect of grain size on the properties of a low nickel austenitic stainless steel. They found that the micro-hardness increased with the decreasing grain size, as did the yield and tensile strength [25]. Li et al. also concluded that the Hall–Petch dependency for both yield strength and tensile strength was valid for the grain size range in the nickel-free high nitrogen austenitic stainless steel [26].

In addition to forming compounds, titanium could be presented as a solid solution state in steel. It is known that Ti was a strong ferrite forming element, and could promote formation of ferrite in austenitic stainless steel or increase ferrite fraction in duplex stainless steel. Jang et al. investigated Cr contents on tensile and corrosion behaviors of 0.13 pct N-containing CD4MCU (0Cr26Ni5Mo2Cu3) cast duplex stainless steels [27]. They found that the tensile behavior of duplex stainless steel increased in a nonlinear manner with increasing Cr content. They believed the major reasons for this trend appeared to be the increase of ferrite fraction and the second precipitates of fine austenitic phase in the ferrite matrix rather than the intrinsic chromium effect. They also investigated different Mo contents on tensile and corrosion behaviors of CD4MCU cast duplex stainless steels [28]. Son et al. investigated N addition on the tensile and corrosion behaviors of CD4MCU cast duplex stainless steels [29]. They concluded that the initial increase in yield strength with N addition was the result of the beneficial shape change of austenitic phase, as well as the intrinsic solid solution hardening effect of N. However, the yield strength decreased with further addition of N owing to the increase in the fraction of austenitic phase, which had inherently lower strength than the ferritic phase. So, it is necessary to investigate the effect of titanium addition on the as-cast structure and mechanical properties of 20Cr-8Ni stainless steel in the laboratory.

In the present work, the as-cast structure and microstructure of ingots with different Ti content were investigated firstly. Moreover, the evolution of typical inclusions in 20Cr-8Ni stainless steel was analyzed. The mechanical property of ingot was detected. On the basis of the calculation result via Thermo-Calc software (version 2017a, Thermo-Calc software, Stockholm, Sweden), the effect of Ti on the as-cast structure and mechanical property of 20Cr-8Ni stainless steel was investigated.

## 2. Materials and Methods

The ingots with different titanium content were produced in the laboratory. The high-purity materials including iron block, chromium blocks, nickel blocks, and so on were firstly melted in a vacuum induction furnace. After those materials were melted completely at 1650 °C, the titanium blocks were added into the melt. For homogenization of the composition, the melt was held for 5 min. Next, the melt was cast into mold when the temperature decreased to 1520 °C. The mold was made of cast iron, and its size is shown in Figure 1a. The ingot was tapped after 5 min and put into the bucket full of water to quench to room temperature. The temperature of the water was around 20 °C. There were three ingots with different Ti content investigated in this work. The contents of carbon, sulfur, nitrogen, and total oxygen were analyzed by the inert gas fusion-infrared absorptiometry method. The contents of silicon, manganese, molybdenum, and titanium were determined by the inductively coupled plasma optical emission spectrometry method. Moreover, the contents of chromium and nickel were detected through spark discharge atomic emission spectrometric analysis. The composition of three ingots is shown in Table 1.

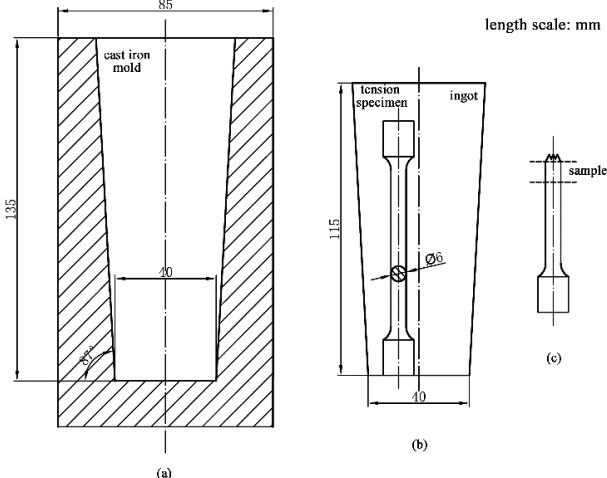

**Figure 1.** (**a**) The size of cast iron mold and the sampling method; (**b**) tensile specimen from ingot; (**c**) sample from tensile specimen after testing.

**Table 1.** Composition of ingots, mass%.

| Ingot | C | Si | Mn | S | Cr | Ni | Mo | Ti | N | O |
|-------|-----|-----|-----|-----|-----|-----|-----|-----|-----|-----|
| S1 | 0.0170 | 1.60 | 1.32 | 0.0080 | 20.11 | 8.55 | 0.4 | 0.0036 | 0.0190 | 0.0056 |
| S2 | 0.0076 | 1.59 | 1.38 | 0.0071 | 20.10 | 8.46 | 0.4 | 0.20 | 0.0170 | 0.0012 |
| S3 | 0.0074 | 1.65 | 1.37 | 0.0077 | 20.15 | 8.58 | 0.4 | 0.45 | 0.0081 | 0.0017 |

The ingot was cut into two parts verticality through the diameter. One part was etched by aqua regia solution to reveal the as-cast structure. The tensile specimen was processed from the other part of ingot shown in Figure 1b. The tensile test was carried out by the electronic universal testing machine at 350 °C. The tensile property at 350 °C is a performance index of castings in industrial practice. Tensile specimens were heated up to the experiment temperature with a heating rate of 10 °C/min. The specimens were pulled to fracture with $1 \times 10^{-3}$ strain rate at the tensile test temperature, followed by air cooling. After the tensile test, the sample was cut near the fracture of the tensile specimen shown in Figure 1c.

The sample from each tensile specimen was polished and etched by mixed solution of 2 g potassium metabisulfite, 20 mL hydrochloric acid, and 50 mL $H_2O$ at 50 °C to reveal the microstructure and grain size of ingot. The microstructure was observed through Leica DM4000M optical microscope (OM, made in Leica, Beijing, China). After that, the grain size and ferrite fraction were analyzed through Image-Pro Plus (version 6.0, Media Cybernetics, Bethesda, MD, USA). The element distribution between ferrite and austenite was detected by electron probe (EPMA, made in Shimadzu, Hong Kong, China). The typical inclusions in samples from each tensile specimen were observed and analyzed through scanning electron microscopy (SEM, made in FEI, Hillsboro, OR, USA) equipped with energy dispersive spectroscopy (EDS). Inclusion statistics was conducted through automatic scanning electron microscope (EVO18-INCAsteel, ZEISS Co. Ltd., Jena, Germany). The maximum diameter of the inclusion was defined as the size of inclusions. Thermo-Calc software (version 2017a, Thermo-Calc software, Stockholm, Sweden) was used to analyze the evolution of typical inclusions and solidification process through Poly and Scheil model.

## 3. Results

### 3.1. As-Cast Structure of Ingots

After etching, the as-cast structures of three ingots are shown in Figure 2. As shown in Figure 2a, the as-cast structure was composed of columnar grains and equiaxed grains in ingot S1 without

titanium addition. The equiaxed grain ratio of ingot S1 was measured to be 51 percent through Image-Pro Plus software. When the titanium content increased to 0.2 mass percent, the equiaxed grain ratio of ingot S2 increased up to nearly 100 percent. The same result was obtained in ingot S3 with the addition of 0.45 mass percent titanium. It could be concluded that titanium addition increased the equiaxed grain ratio of ingot.

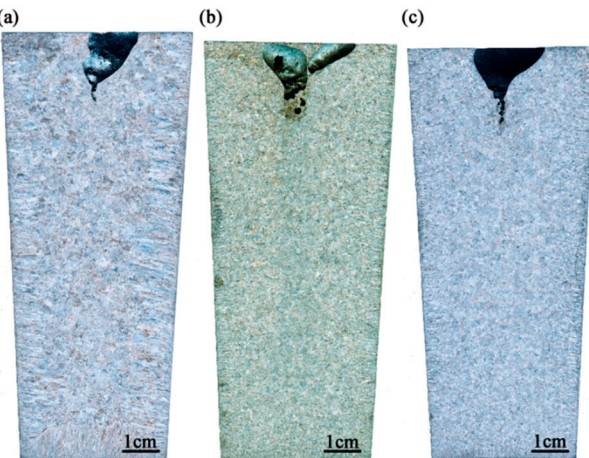

**Figure 2.** As-cast structure of three ingots: (**a**) S1 with no Ti addition; (**b**) S2 with 0.2 mass% Ti; and (**c**) S3 with 0.45 mass% Ti (online version in color).

After polishing and etching, the microstructures of three samples from each tensile specimen after 350 °C tensile testing with low magnification are shown in Figure 3. The austenite located on the grain boundary (BA) or in the grain interior (IA). The microstructures of three samples were all composed of equiaxed grains owing to the sampling location of the tensile specimen in ingots shown in Figure 1b. However, the grains in S1 were obviously larger than S2 and S3. After statistics through Image-Pro Plus, the results about grain size and density are shown in Figures 4 and 5.

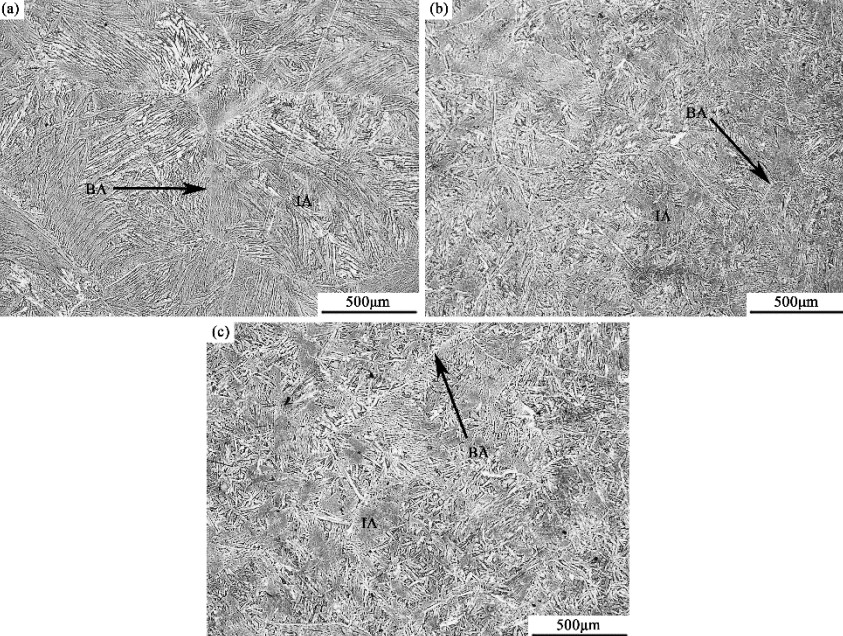

**Figure 3.** Microstructure of three samples with low magnification: (**a**) S1; (**b**) S2; and (**c**) S3.

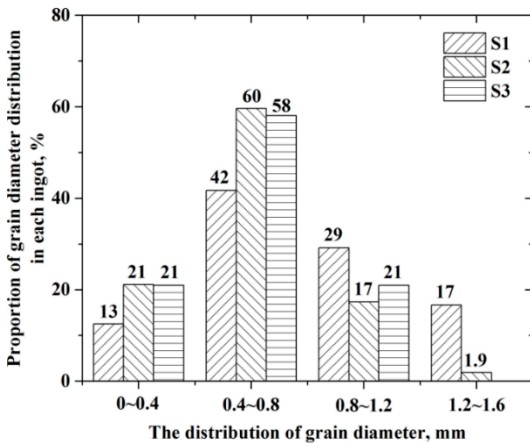

**Figure 4.** Grain diameter distribution of each sample.

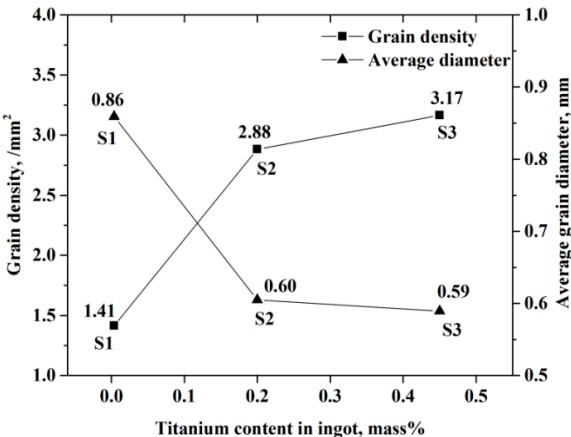

**Figure 5.** Grain quantity of each sample.

Figure 4 shows the grain diameter distribution of the three samples. The grain diameter was divided into four groups, including 0–0.4 mm, 0.4–0.8 mm, 0.8–1.2 mm, and 1.2–1.6 mm. Compared with S1, the proportion of grains smaller than 0.8 mm increased, and the proportion of grains larger than 0.8 mm decreased in S2. The distribution of grains' diameter in S3 was similar to that in S2. Figure 5 shows the grain density (the number of grains per unit area) and average grain diameter of the three samples. It was obvious that the grain density in S2 was larger than in S1, and the average grain diameter in S2 was smaller than in S1. The results in S3 were similar to those in S2. The conclusion could be drawn that the size of grains decreased when the titanium content increased up to 0.2 mass percent, but the grain size decreased little with the further increase of titanium.

### 3.2. Microstructure and Element Distribution

The microstructures of the three samples from each tensile specimen with high magnification are shown in Figure 6. The bright phase was austenite, and the grey phase was ferrite. The element mapping of austenite and ferrite is shown in Figure 7. It was obvious that ferrite was rich in chromium, but lacked nickel. Figure 6a,c,e shows the austenite on the grain boundary, and Figure 6b,d,f shows the austenite in the grain interior. This figure clearly demonstrates that the microstructure of 20Cr-8Ni stainless steel varied significantly with different titanium contents. As shown in Figure 6a,b, the size of austenite was small in S1. With increasing titanium content up to 0.2 mass percent, the size of austenite and the mass fraction of ferrite increased. With the further addition of titanium up to 0.45 mass percent, the morphology of austenite changed little, but the mass fraction of ferrite increased to a higher degree. After statistics through at least 25 fields in each sample, the mass fraction of ferrite

in samples with different titanium contents is shown in Figure 8. It was obvious that the ferrite mass fraction increased with the increasing titanium content. Moreover, the mass fraction of ferrite could increase up to 42.6 percent with the increase of titanium content to 0.45%.

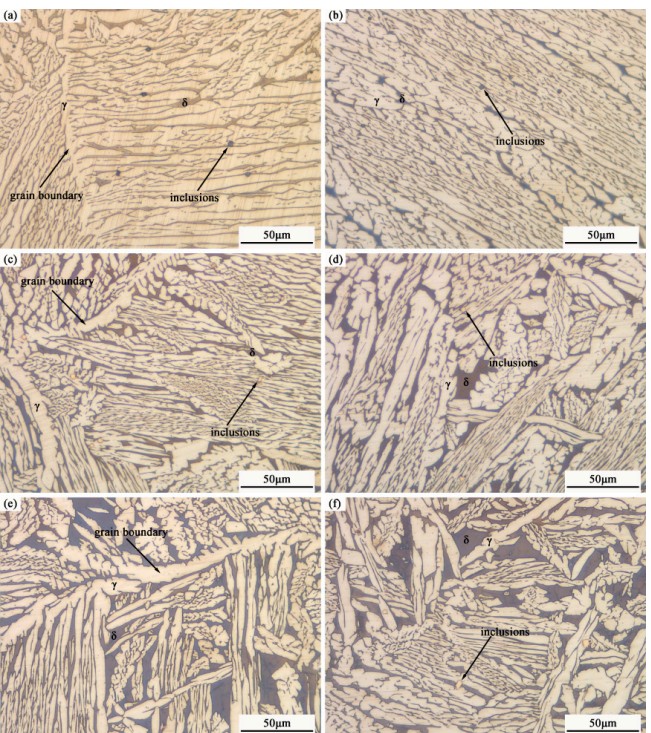

**Figure 6.** Microstructure of the three samples with high magnification: (**a**,**b**) S1; (**c**,**d**) S2; and (**e**,**f**) S3 (online version in color).

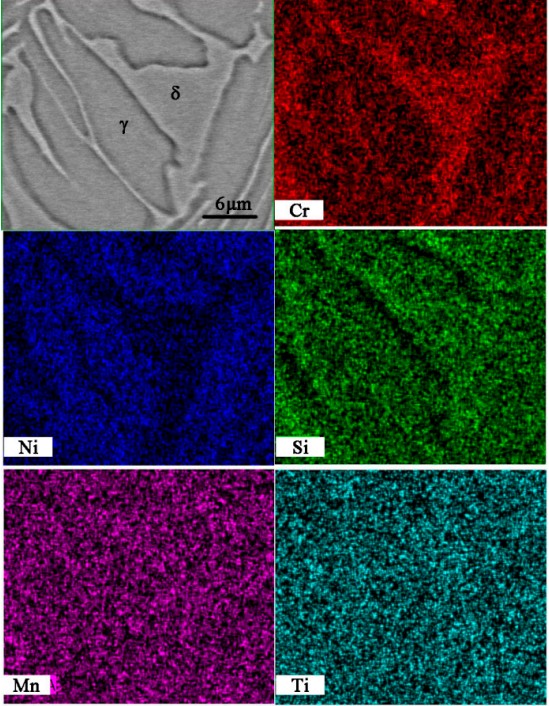

**Figure 7.** Element mapping of ferrite and austenite by energy dispersive spectroscopy (EDS) (online version in color).

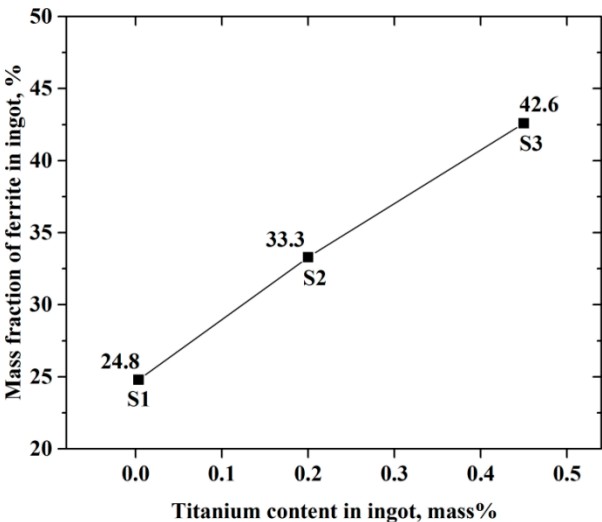

**Figure 8.** Mass fraction of ferrite in samples with different Ti content.

After point detection by EPMA, the titanium distribution between ferrite and austenite is shown in Figure 9. The detected titanium content in ferrite and austenite increased with titanium addition in ingot. Besides, the titanium content in S1 was low, which made no difference between titanium content in ferrite and austenite. In S2 and S3, the titanium content in ferrite was higher than in austenite.

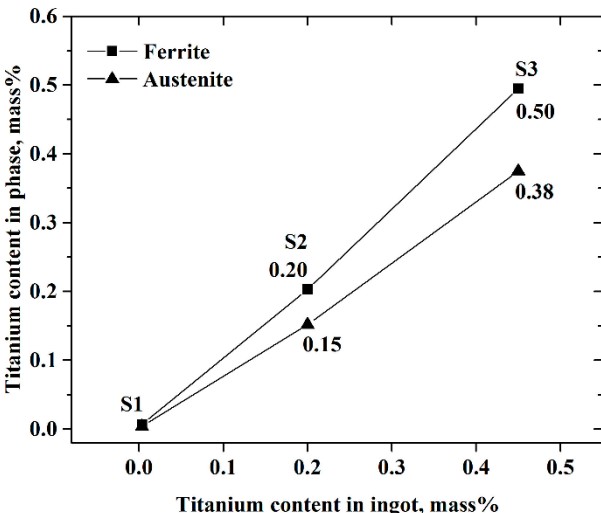

**Figure 9.** Titanium distribution in ferrite and austenite by electron probe (EPMA).

### 3.3. Typical Inclusions

As shown in Figure 6, the inclusions in ingot S1 were black and spherical, which were speculated as oxides. After titanium addition, the inclusions in ingots S2 and S3 were rectangular and orange TiN through OM. The inclusions were mostly located in grain interior. Meanwhile, they distributed in austenite, ferrite, or the interface of two phases. In order to investigate the influence of titanium contents on inclusions, the inclusions in samples from each tensile specimen were observed and analysed through SEM and EDS. The morphology and composition of typical inclusions are shown in Figure 10 (the numbers beside the inclusion are atomic percentages determined by EDS analysis).

As shown in Figure 10A1,A2, the typical inclusions in S1 without titanium addition were spherical or subspherical dual phase oxides. On the basis of the morphology of oxides, it could be speculated that those oxides were liquid at melting temperature. With temperature decreasing during the casting

process, the oxides changed from liquid to solid gradually. Finally, the dual phase oxides with different compositions formed in ingots. On the basis of EDS analysis, the grey area was oxide-rich in titanium and manganese, and the black area was oxide-rich in silicon. The element mapping of a typical dual phase oxide is shown in Figure 11a. The elements' distribution was consistent with results shown in Figure 10A1,A2.

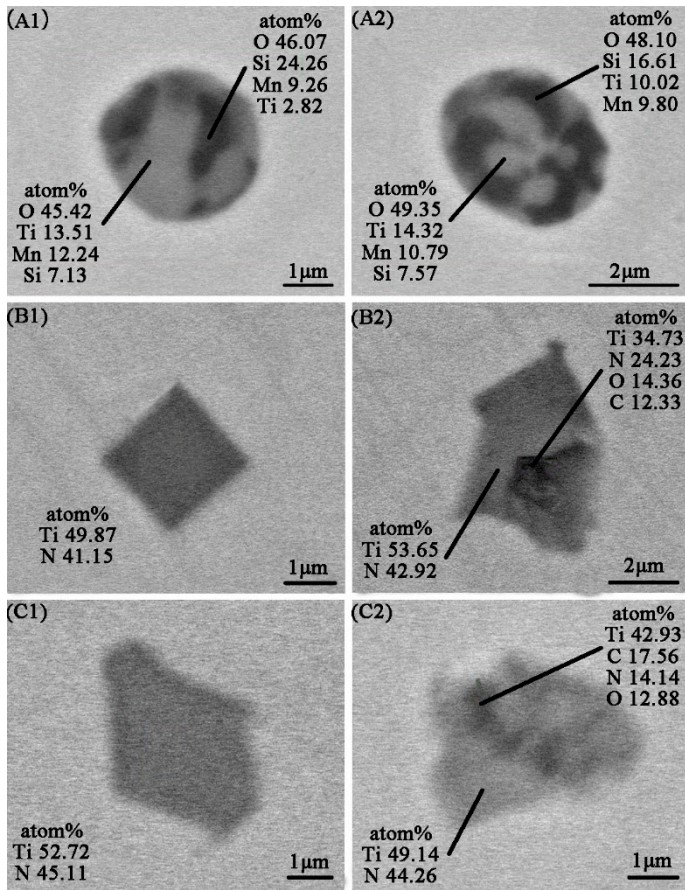

**Figure 10.** Morphology and type of typical inclusions in samples: (**A1**,**A2**) S1; (**B1**,**B2**) S2; and (**C1**,**C2**) S3.

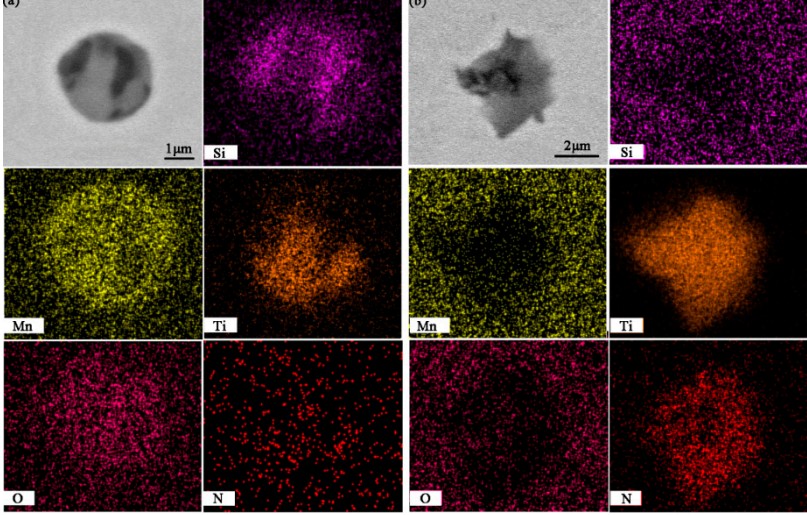

**Figure 11.** Elemental mapping of (**a**) a typical dual phase oxide formed in S1 condition; (**b**) a typical complex TiN formed in S2 (online version in color).

With titanium content increasing up to 0.20 mass percent, the typical inclusion in ingot S2 was pure TiN, as shown in Figure 10B1. Besides, there were also complex inclusions observed in ingot S2, as shown in Figure 10B2. On the basis of EDS analysis, the grey area was TiN, and the black area was titanium oxide owing to the higher content of oxygen. The element mapping of a typical dual phase inclusion is shown in Figure 11b. The black area lacked nitrogen, but was rich in oxygen, which was consistent with result shown in Figure 10B2. With further addition of titanium up to 0.45 mass percent, the typical inclusions in S3 were similar to those in S2, as shown in Figure 10C1,C2.

After statistics, the inclusion size and density of each samples are shown in Figure 12. The inclusions in ingot S1 were mainly oxides rich in Si, Mn, and Ti. The density of oxides was about 123/mm$^2$. Nitride was the main inclusion in ingot S2 and S3, and the densities were about 195 and 208/mm$^2$, separately.

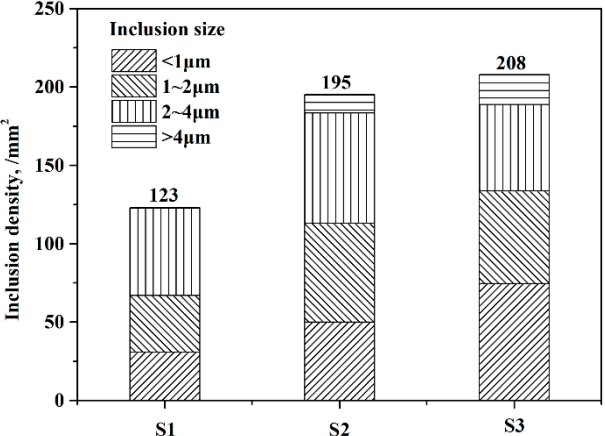

**Figure 12.** Inclusion size and density of each sample.

### 3.4. Tensile Properties

Table 2 shows the effect of titanium content on the tensile property of 20Cr-8Ni stainless steel at 350 °C. Comparing ingot S2 with S1, the yield strength (YS) value increased from 207 MPa to 229 MPa. With the titanium content further increasing up to 0.45 mass percent, the YS value increased to 238 MPa. Similarly, the ultimate tensile strength (UTS) value increased gradually with the increasing titanium content. However, the tensile elongation changed little with the increasing titanium content.

**Table 2.** Tensile properties of ingots with different Ti content at 350 °C. YS, yield strength; UTS, ultimate tensile strength.

| Ingot | Ti Contents (mass%) | YS (MPa) | UTS (MPa) | Tensile Elongation (%) |
|-------|---------------------|----------|-----------|------------------------|
| S1 | 0.0036 | 207.41 | 452.31 | 31.20 |
| S2 | 0.20 | 229.11 | 471.24 | 30.92 |
| S3 | 0.45 | 238.89 | 494.87 | 31.20 |

## 4. Discussion

### 4.1. Formation of TiN and Effect on As-Cast Structure

To investigate the effect of titanium on the inclusions and as-cast structure, the equilibrium solidification process of ingots with different titanium content was calculated through Thermo-Calc software. The composition of ingots is shown in Table 1. The calculated temperature ranged from 1000 °C to 1600 °C, and the calculated results are shown in Figure 13.

Those figures showed that the inclusions during solidification process included oxides, TiN, MnS, and Ti$_4$C$_2$S$_2$. MnS and Ti$_4$C$_2$S$_2$ would not be investigated in this article. As shown in Figure 13a, the typical oxides were Si-Mn-Ti-O in ingot S1, which was consistent with the results shown in Figure 10A1,A2 and Figure 11a. Seo et al. also found MnSiO$_3$ inclusions and (Mn, Ti)-spinel oxide in

Ti-containing steel weld metals [12]. TiN would form at 1300 °C when the ingot solidified completely, but was not observed in ingot S1.

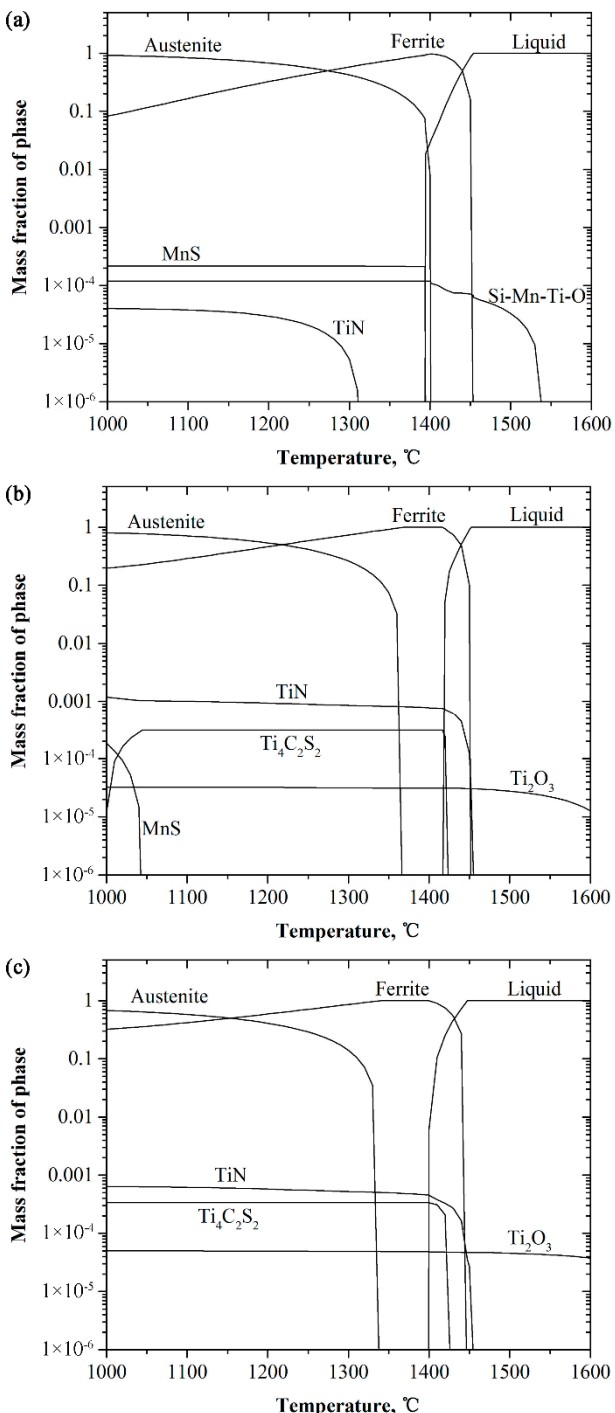

**Figure 13.** Equilibrium thermodynamic analysis in the solidification process of 20Cr-8Ni stainless steel: (**a**) S1; (**b**) S2; and (**c**) S3.

With titanium content increasing up to 0.20 mass percent, the calculated result was shown in Figure 13b. The calculated oxide changed to $Ti_2O_3$, and TiN formed during cooling and solidification process. The pure $Ti_2O_3$ was not observed in ingot S2. However, the complex $Ti_2O_3$ + TiN inclusion was detected shown in Figures 10B2 and 11b. On the basis of Hou's research, the disregistry between

TiN and Ti$_2$O$_3$ was 13.61, and Ti$_2$O$_3$ could promote the formation of TiN [30]. In ingot S3, the calculated inclusions were similar with the results in ingot S2.

Bramfitt indicated TiN was very effective as a nucleating agent for δ-Fe based on misfit calculation results [18]. Park concluded TiN and MgAl$_2$O$_4$-TiN complex inclusions promoted the formation of fine equiaxed grains when Ti was added in ferritic stainless steel [19]. Hou found the Ti$_2$O$_3$-TiN complex nucleus increased equiaxed grain ratio of Ti bearing Fe-18Cr ferritic stainless steel [30,31]. To evaluate the effect of TiN and Ti$_2$O$_3$-TiN on the as-cast structure of 20Cr-8Ni stainless steel, the scheil solidification process was calculated. The result is shown in Figure 14. The formation of oxides was not taken into consideration during the scheil solidification process. As shown in Figure 14a, TiN formed at the end of solidification process in ingot S1 owing to element segregation, which was different from Figure 13a. Similarly to Figure 13b,c, TiN and Ti$_2$O$_3$-TiN complex inclusions formed before and during the solidification process in ingot S2 and S3, as shown in Figure 14b,c. As a result, the heterogeneous nucleation did not happen in ingot S1. In ingot S2, the ratio of equiaxed grains was nearly 100 percent, as shown in Figure 2b, owing to heterogeneous nucleation. Besides, the size of equiaxed grains in ingot S2 was smaller than S1 shown in Figures 3 and 5. Villafuerte et al. also found the size of equiaxed grains decreased with increasing titanium contents above 0.18 mass percent in full penetration gas-tungsten arc welds on ferritic stainless steel plates [32]. In ingot S3, the solidification process was similar to ingot S2. The equiaxed zone ratio was also nearly 100 percent, but the equiaxed grains size changed little compared with S2. Although the titanium content in ingot S3 was higher than in ingot S2, the nitride densities in S2 and S3 were the same, as shown in Figure 12. Moreover, the mass fraction of TiN in S3 was nearly the same as S2, as shown in Figure 13. This was caused by the lower content of nitrogen in ingot S3 than in ingot S2.

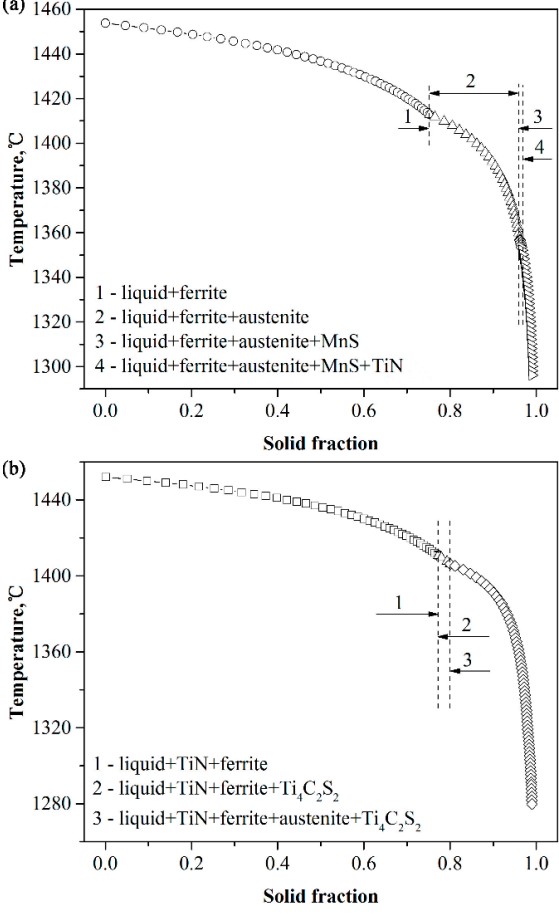

**Figure 14.** *Cont.*

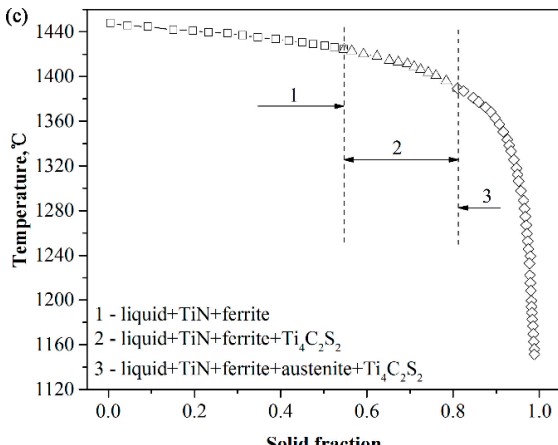

**Figure 14.** Scheil solidification calculation result for 20Cr-8Ni stainless steel: (**a**) S1; (**b**) S2; and (**c**) S3.

### 4.2. Effect of Titanium on Microstructure

To evaluate the effect of titanium content on the microstructure of 20Cr-8Ni stainless steel, it was necessary to discuss the formation of auetenite durnig the solidification process or after the solidification process. On the basis of the equilibrium solidification process shown in Figure 13, the austenite formed through peritectic or eutectic reaction at the end of solidification process in ingot S1. The equilibrium solidification model of ingot S1 was the FA model (liquid → liquid + δ-ferrite → liquid + δ-ferrite + γ-austenite → δ-ferrite + γ-austenite). Liang et al. also indicated that the equilibrium solidification model of AISI 321 stainless steel was the FA model, according to the ratio of Ni equivalent and Cr equivalent [33]. In ingot S2 and ingot S3, the austenite formed after solidification, and the equilibrium solidification model was the F model (liquid → liquid + δ-ferrite → δ-ferrite). During the scheil solidification process shown in Figure 14, the austenite in ingot S1, S2, and S3 formed during solidification owing to element segregation. However, the solid fraction at which austenite precipitated increased with titanium addition. It could be concluded that titanium prevented the precipitation of austenite in 20Cr-8Ni stainless steel.

According to Hammar and Svensson equivalents shown in Equations (1) and (2), the solidification models of ingot S1, S2, and S3 were all the F model [34]. This meant that equilibrium solidification calculated through Thermo-Calc software was more reliable. On the basis of the equilibrium solidification process, the soluble titanium content in ferrite and austenite at different temperatures was calculated. The results are shown in Figure 15. Figure 15a shows the soluble titanium content in ferrite when austenite began to form. $T_A$ was the temperature at which austenite began to form. With soluble titanium content in ferrite increasing through S1, S2, and S3, the stability of ferrite increased gradually. Moreover, transformation of ferrite to austenite retarded. Figure 15b shows the soluble titanium content in ferrite and austenite at 1100 °C. The solution treatment temperature for 20Cr-8Ni stainless steel in industrial practice was around 1100 °C. The content of soluble titanium in ferrite increased through S1, S2, and S3, and was higher than austenite, which was consistent with the EPMA result shown in Figure 9. Figure 16 showed the calculated ferrite fraction at 1100 °C, and the calculated result was consistent with the experiment one.

$$Cr_{eq} = (\%Cr) + 1.37(\%Mo) + 1.5(\%Si) + 2(\%Nb) + 3(\%Ti) \tag{1}$$

$$Ni_{eq} = (\%Ni) + 22(\%C) + 14.2(\%N) + 0.31(\%Mn) + (\%Cu) \tag{2}$$

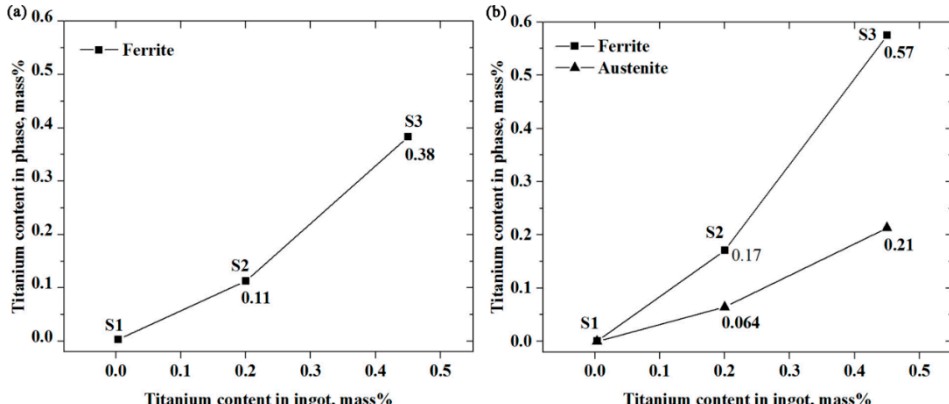

**Figure 15.** Calculated Ti distribution in phase at different temperature through equilibrium thermodynamic analysis: (**a**) $T_A$ and (**b**) 1100 °C

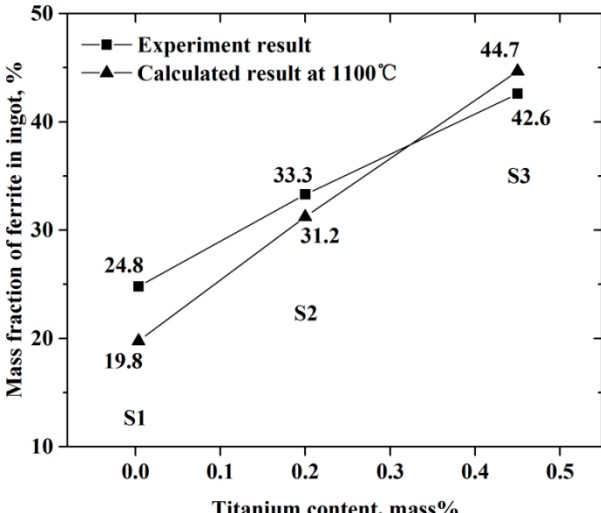

**Figure 16.** Experiment mass fraction of ferrite in ingots and the calculated result through equilibrium thermodynamic analysis.

*4.3. Improvement of Tensile Property*

On the basis of the discussion above, the mechanism for tensile property improvement would be discussed in this part. Comparing S1 with S2 or S3, the smaller equiaxed grain size and higher mass fraction of ferrite improved the YS and UTS. Comparing S2 with S3, the higher mass fraction of ferrite caused the higher YS and UTS, although the equiaxed grain size was nearly the same. Jang and Son et al. investigated the effect of microstructure on tensile property in CD4MCU cast duplex stainless steels [27–29]. They concluded that the YS and UTS increased with increasing mass fraction of ferritic phase, because the ferritic phase had inherently higher strength than the austenitic phase. Besides, the fine equiaxed grains also improved the tensile property, which was similar to Schino and Li's research [25,26].

**5. Conclusions**

To refine the solidification structure of 20Cr-8Ni stainless steel for heavy castings in industrial practice, the effect of titanium addition on the as-cast structure and high-temperature tensile property was investigated in the laboratory. According to the analysis of the as-cast structure, microstructure, inclusions type, and tensile properties of ingot with different titanium content, the following conclusions were obtained:

(1)    With the increasing titanium content, the typical inclusions changed from dual phase Si-Mn-Ti oxides to pure TiN or complex $Ti_2O_3$ + TiN. Meanwhile, the equiaxed grain ratio of ingot increased from 51 percent to nearly 100 percent, and the size of equiaxed grain decreased owing to heterogeneous nucleation of δ-Fe;

(2)    With the increasing titanium content, the soluble titanium content in ferrite phase increased, retarded the formation of austenite phase, and changed the equilibrium solidification model from FA to F. As a result, the mass fraction of ferrite phase in ingot increased gradually;

(3)    Owing to increase of the ferrite phase mass fraction and decrease of the equiaxed grain size, the YS and UTS at 350 °C of ingot increased gradually, but tensile elongation changed little.

**Author Contributions:** Writing—review and editing, Q.W.; writing—original draft preparation, Q.W.; conceptualization, G.C.; formal analysis, Y.H. All authors have read and agreed to the published version of the manuscript.

**Funding:** This work was supported by the National Natural Science Foundation of China (No. 51674024).

**Acknowledgments:** The authors would like to appreciate NCS Testing Technology Co., Ltd. for contributing to the testing and inspection.

**Conflicts of Interest:** The authors declare no conflict of interest.

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
