# Peer review of "Effect of Titanium Addition on As-Cast Structure and High-Temperature Tensile Property of 20Cr-8Ni Stainless Steel for Heavy Castings"

_metals, doi:10.3390/met10040529_

Round 1

Reviewer 1 Report

Points in favor:

The present work has been performed systematically and the influence of Ti addition to the as-cast structure of 20Cr-8Ni steel was well correlated with microstructure investigations by SEM and EPMA.

Points Detracting:

  1. Please write out the abbreviation CD4MCU in line 61 on page 2 for better clarity. What does this designation stand for?
  2. What is meant by chromium blocks in line 80 and titanium blocks in line 83 on page 2?
  3. What is the length scale of the mold given in Figure 1? cm, mm?
  4. The authors talk about heavy castings, but in the paper the mass of an ingot is about 1.3 kg, when the dimensions are given in mm in Fig. 1!
  5. How was the chemical composition (Table1) of the ingots determined? Please include the information in the chapter Materials and Methods.
  6. Why were the tensile samples tested at 350 °C and not at RT? What was the purpose to use this temperature?
  7. On page 3, in line 94 the word tension should be corrected to tensile due to performing a tensile test. This mistake was also made in some other parts of the paper.
  8. Please include additional information for EPMA and SEM in the chapter Materials and Methods, for example company of the devices, operating parameters and so on.
  9. Which database was used for the Thermo-Calc calculations? Add this information in the chapter Materials and Methods, too.
  10. How was the Ti content in solid solution determined in Figure 8?
  11. For which alloy was the determination of oxides and TiN by elemental mapping carried out?
  12. With a ferrite content of 24.8 vol.% (without Titanium) up to 44.7 vol.% ferrite (0.45wt.% Ti) in the cast structure, it can still be called an austenitic steel? Is it not a duplex steel, especially for 0,45 wt.% Ti?

Reviewer 2 Report

The subject matter of the article corresponds to the thematic sections of the journal.
The results correspond to the goals and objectives set in the work. Presentation of the results needs a little adjustment.
Notes on the results section.
1. Figures 2 and 3 are not too informative. The structural features of the samples are barely distinguishable.
Give the image of A2, B2 and C2 in Figure 3 at a higher magnification.
2. The term grain density is not understood. It needs to be clarified. 5.
3. A typo on line 148. There should be 20Cr-8Ni, not 28Cr-8Ni.
The discussion and conclusions confirm the experimental results and theoretical calculations, which increase the significance of this work.
In general, the work is not innovative, but presents interesting experimental results.

Reviewer 3 Report

This study evaluates the influence of the Ti addition on the microstructure and tensile properties (at 350 ºC) of the as cast 20Cr-8Ni alloy. These authors discussed the influence of inclusions on the grain structure and phase evolutions formed during casting. They also verified the experiments with the simulation results. However, this manuscript requires some improvements to reach an acceptable quality level for publication as in the following:

  • According to the scales you provided in Figures 2 and 3, your samples do not represent heavy castings. You also did not provide conditions to stimulate a cooling rate as small as of a large section casting.
  • Regarding your innovation, it should be better elaborated in the manuscript!
  • Since you discuss the inclusion formation, in correlation with the melting temperature, you should present the melting temperature in lines 81 and 82.
  • What technique did you use to identify the chemical compositions presented in Table 1?
  • Please provide similar decimals in Table 1.
  • What is the unit of dimensions mentioned in Figure 1?
  • Is it possible to increase the quality of three images illustrated in Figure 2?
  • In Figure you need to balance the proportions because:
    • 13+42+29+17 = 101
    • 21+60+17+1.9 = 99.9
  • Could you please provide the formula you calculated the grain densities presented in Figure 5?
  • Please provide the standard deviations of average grain diameters presented in Figure 5.
  • Which sample (S1, S2 or S3) was illustrated in Figure 7?
  • In the caption of Figure 11, it is better to specify some details: (a) a typical dual phase oxide formed in S1 condition; (b) an inclusion mainly composed of TiN formed in S2.
  • Table 2 shows the results of how many tensile tests for each condition? Please provide similar decimals in Table 2.
  • Citation is missing in line 214.
  • Please correct the format of citation in line 222.
  • Citations in lines 227 and 228 are missing.
  • Since in your simulations, the results have indicated the influence of S on the formation of inclusions, it is better to revise your EDS results and maps in order to add the S values.
  • Equation 1 in figure 14 is not correct.
  • The text needs to be revised to correct some typing errors (e.g., modelaccording - line 255, Thermo-Cala - line 208).
  • The pages in some references are incorrect or missing (e.g., references [31],[32], and [34])

Reviewer 4 Report

Effect of Ti addition on the refinement of as-cast structure in 20Cr-8Ni austenitic stainless steel was studied in this manuscript. The effect of TiN as a heterogeneous nucleation site of d-ferrite has been reported in many previous researches, but the Ti-content-dependent behavior can be an informative point in this manuscript. Overall discussion on the results needs to be revised. The reviewer suggests several corrections as follow;

  1. In the title, a high-temperature tensile property is not matched with the contents. 350° is not high temperature. Why authors select and use the temperature for the tensile test? Need to explain it.
  2. Change of type of inclusion by Ti addition was introduced in the manuscript. Fraction or number density should be added to the result.
  3. Nitrogen contents are decreasing as Ti addition increases. Considering the increased TiN fraction and matrix Ti contents where Ti shows strong bonding response, the decrease of nitrogen contents is odd. The explanation is necessary because the authors used the decreased nitrogen contents for the discussion on the TiN formation behavior in higher Ti addition alloy (S3).

    4. Comparing Fig. 9 and Fig. 14, the tendency of titanium content in each phase is different.

In the case of S3, the observed Ti content in ferrite is lower than the calculated one. However, Ti content in austenite is the opposite. Is there any reason?

Round 2

Reviewer 4 Report

The revised version of manuscript is enough to be published.

Author Response

Thanks for your comments.